# Low blank sampling method for measurement of the nitrogen isotopic composition of atmospheric NO$_x$

**Kazuki Kamezaki**[1]*, **Takahisa Maeda**[1], **Shigeyuki Ishidoya**[1], **Ayumi Tsukasaki**[1], **Shohei Murayama**[1], **Naoki Kaneyasu**[1,2]

1 Environmental Management Research Institute, National Institute of Advanced Industrial Science and Technology (EMRI/AIST), Tsukuba, Japan, 2 Fukushima Institute for Research, Education and Innovation, Namie-machi, Fukushima, Japan

* kamezaki-k@aist.go.jp

**Data Availability Statement:** The datasets generated and analyzed during this study are available via Zenodo at https://doi.org/10.5281/zenodo.10398614.

## Abstract

The nitrogen isotopic composition of nitrogen oxide (NO$_x$) is useful for estimating its sources and sinks. Several methods have been developed to convert atmospheric nitric oxide (NO) and/or nitrogen dioxide (NO$_2$) to nitrites and/or nitrates for collection. However, the collection efficiency and blanks are poorly evaluated for many collection methods. Here, we present a method for collecting ambient NO$_x$ (NO and NO$_2$ simultaneously) with over 90% efficiency collection of NO$_x$ and low blank (approximately 0.5 µM) using a 3 wt% hydrogen peroxide (H$_2$O$_2$) and 0.5 M sodium hydride (NaOH) solution. The 1σ uncertainty of the nitrogen isotopic composition was ± 1.2 ‰. The advantages of this method include its portability, simplicity, and the ability to collect the required amount of sample to analyze the nitrogen isotopic composition of ambient NO$_x$ in a short period of time. Using this method, we observed the nitrogen isotopic compositions of NO$_x$ at the Tsukuba and Yoyogi sites in Japan. The averaged δ$^{15}$N(NO$_x$) value and standard deviation (1σ) in the Yoyogi site was (−2.7 ± 1.8) ‰ and in the Tsukuba site was (−1.7 ± 0.9) ‰ during the sampling period. The main NO$_x$ source appears to be the vehicle exhaust in the two sites.

## Introduction

Nitric oxide (NO) and nitrogen dioxide (NO$_2$) are collectively referred to as nitrogen oxide (NO$_x$). NO$_x$ is a primary pollutant in the atmosphere and is involved in urban environmental issues such as photochemical smog, acid rain, tropospheric ozone production, and human health. Besides, NO$_x$ deposition can enhance ecosystem productivity through fertilization or decrease it through nutrient imbalances and reduce ecosystem biodiversity through acidification and eutrophication [1]. NO$_2$ is oxidized to nitrate (NO$_3{}^-$), which is adsorbed by aerosols and transported over long distances, affecting distant environments [2]. In areas affected by human pollution, fossil-fuel combustion from traffic, residential heating, cooking, industry, and energy sectors are the main sources of NO$_x$. On the other hand, as a natural NO$_x$ source, biomass burning, biogenic production, and lightning are also important sources of NO$_x$ [3].

**Funding:** This study was supported by the Steel Foundation for Environmental Protection Technology in the form of a grant to KK [FY2021–2022] and by the Japan Society for the Promotion of Science (JSPS) in the form of Grants-in-Aid for Scientific Research (KAKENHI Program) to SI [JP22H05006], and SI and KK [JP22H05006].

**Competing interests:** The authors have declared that no competing interests exist.

Global annual $NO_x$ emissions have been gradually curtailed [4]. However, it is important to understand the exact behaviour of $NO_x$ to elucidate how the suppression of $NO_x$ emissions changes atmospheric reactions.

The nitrogen isotopic composition ($\delta^{15}N$ value) of $NO_x$ is a useful tool for estimating its sources because the nitrogen isotope composition of each source has a unique value (S1 Table). Nitrogen source is identified from the nitrogen isotopic composition of $NO_3^-$ in the aerosol [5–7]. To date, several methods have been developed to convert atmospheric NO and/ or $NO_2$ to nitrites and/or nitrates for collection. A denuder system, filter pack, and Ogawa sampler have all been used to collect ambient $NO_2$ with reagents, such as triethanolamine, guaiacol, and potassium hydroxide [8–13]. A wet method was used to collect $NO_x$ by passing air containing $NO_x$ through the recovery solution. Potassium permanganate ($KMnO_4$) with sodium hydroxide (NaOH) or 20% triethanolamine in water have been used as the recovery solutions for the collection of ambient $NO_x$ [14–16]. In addition, a solid sorbent method with attached chemical reagents has also been reported for the collection of ambient $NO_x$ [17]. It is important that the collection efficiency of $NO_x$ is close to 100% and that the blank is small for the isotopic composition analysis. However, the collection efficiency and blanks are poorly evaluated for many collection methods and the locations at which the $\delta^{15}N$ values of $NO_x$ were measured were limited owing to the difficulty of the measurement method.

Recently, a high-time-resolution method for $NO_x$ collection was developed using gas-washed bottles in $KMnO_4$ and NaOH recovery solutions. This method shows high collection efficiency for $NO_x$, whereas a high concentration of $NO_x$ blank (approximately 5 μM) is observed [14, 15]. Therefore, at present, there is almost no fully validated simple method that can collect NO and $NO_2$ for the analysis of the $\delta^{15}N$ values of $NO_x$ with high efficiency. Compared to the $KMnO_4$/NaOH recovery solution, the hydrogen peroxide ($H_2O_2$)/NaOH recovery solution can remove $NO_x$ more efficiently, as reported by Ohta et al. [18] and Kuropka, [19].

In highly alkaline conditions, $H_2O_2$ produces various intermediate products that act as oxidants with $H_2O_2$ decomposition [20]. Free radicals generated due to $H_2O_2$ decomposition efficiently oxidize NO. It has been pointed out that particularly oxygen anions ($O_2^-$) produced at high pH may effectively oxidize NO [21]. On the other hand, NO and $NO_2$ dissolve in NaOH solution, and the presence of $H_2O_2$ accelerates the oxidation of $NO_2$ [18, 22]. The mechanism of the reaction of $NO_x$ with $H_2O_2$/NaOH is expressed as follows:

$$H_2O_2 \leftrightarrows OOH^- + H^+ \tag{1}$$

$$H_2O_2 + OH^- \leftrightarrows HOO^- + H_2O \tag{2}$$

$$H_2O_2 + HOO^- \rightarrow OH\cdot + O_2^-\cdot + H_2O \tag{3}$$

$$O_2^-\cdot + NO \rightarrow ONOO^- \tag{4}$$

$$2NO_2 + 2OH^- \rightarrow NO_2^- + NO_3^- + H_2O \tag{5}$$

$$NO_2^- + H_2O_2 \rightarrow NO_3^- + H_2O \tag{6}$$

In this study, we tested and developed a more efficient NO and $NO_2$ collection method using a $H_2O_2$ /NaOH recovery solution for the sampling method and measurement of nitrogen isotopic composition of atmospheric $NO_x$. This method has high $NO_x$ collection efficiency and low $NO_2^-$ and $NO_3^-$ blanks in the recovery solution.

## Materials and methods

### Commercial NO$_x$ samples and recovery solution preparation

Commercial cylinders containing 91 ppm NO (Sample A, Japan Fine Products Co. Ltd., Kanagawa, Japan) and 5 ppm NO$_2$ (Sample B, Japan Fine Products Co. Ltd., Kanagawa, Japan) balanced with N$_2$ were used in this study. To dilute these high-concentration NO and NO$_2$ gases, pure N$_2$ (99.995% purity) was used.

To prepare 200 mL of the recovery solution, a highly concentrated 10 M NaOH solution was prepared using reagent-grade NaOH (Special Grade; FUJIFILM Wako Pure Chemical Corp., Osaka, Japan). 10 mL of concentrated NaOH solution was added and diluted with 80 mL of 18.2 MΩ cm water produced by IQ7010 (Merck Millipore Corporation, Massachusetts, United States) in a beaker. Next, 20 mL of 35 wt% H$_2$O$_2$ (Special grade, FUJIFILM Wako Pure Chemical Corp., Osaka, Japan) was added. The entire solution was made up to 200 mL with an additional 18.2 MΩ·cm water. NaOH was diluted before adding H$_2$O$_2$, because H$_2$O$_2$ decomposes rapidly in highly concentrated basic solutions [23]. The old reagents were not used because H$_2$O$_2$ gradually decomposes; therefore, the prepared reagents were used within one day. In our experimental study, it was observed that refrigerated H$_2$O$_2$ remained usable for a period of six months following its purchase. However, after a duration of nine months, the H$_2$O$_2$ failed to generate bubbles even upon NaOH addition, and its NO$_x$ trapping efficiency was low. After preparing the reagent, the reaction was allowed to proceed for 30 min to 1 h before the H$_2$O$_2$/NaOH recovery solution was used (35 wt% H$_2$O$_2$ was used in this study).

For comparison with the conventional method, we prepared a KMnO$_4$ /NaOH recovery solution with reference to Fibiger et al. [14]. Briefly, 1 N (0.2 M) KMnO$_4$ was prepared from reagent-grade KMnO$_4$ (Special grade, FUJIFILM Wako Pure Chemical Corp., Osaka, Japan). The 125 mL, 1 N KMnO$_4$ was diluted to 300 mL, and 25 mL of 10 M NaOH was added. The entire solution was 500 mL with an additional 18.2 MΩ·cm of water. The KMnO$_4$/NaOH recovery solution was stored in a 500 mL amber glass bottle and used within one day to prevent contamination with NO$_x$.

### Sampling system

A schematic of the sampling system is shown in Fig 1. All the tubes were 1/4-inch. Three bubblers (080100–02, SIBATA, Tokyo, Japan) were used to maintain high NO$_x$ collection

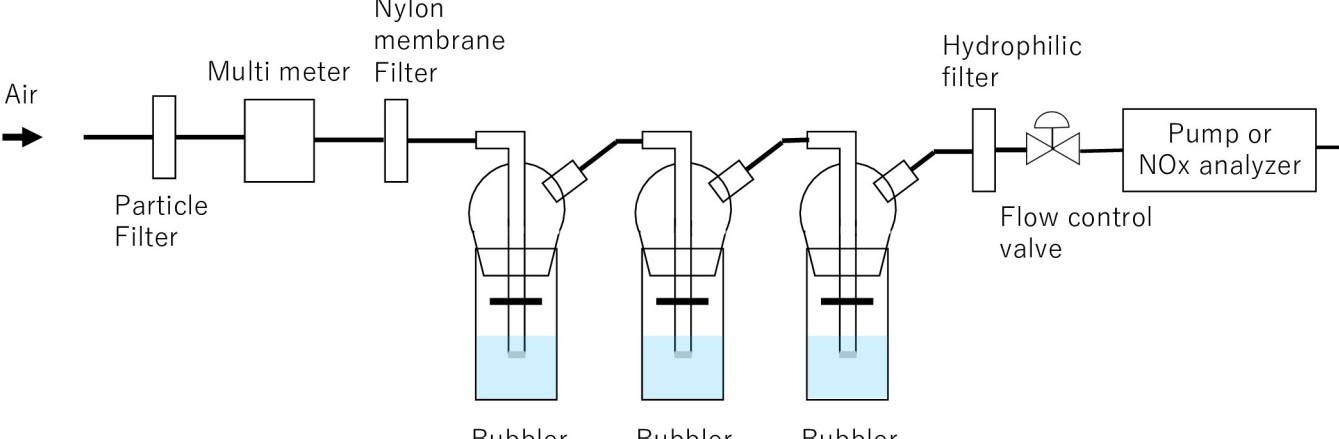

**Fig 1. Schematic diagram of the NO$_x$ sampling system.** When the laboratory experiment was to collect diluted samples A and B, the multi-meter was moved in front of the particle filter.

efficiency. A rubber tube was used as a bubbler joint. A PTFE filter (Advantec Co. Ltd., Tokyo, Japan) and a 0.45 μm pore size Whatman's nylon membrane filter equipped with PFA filter folders (Savillex, Minnesota, United States) were used in front of the bubblers to remove aerosols and gas-phase nitrate ($HNO_3$) from the atmosphere. The flow rate was controlled by using a valve immediately prior to the pump. An 8 μm pore size hydrophilic filter (Merck Millipore Ltd., Massachusetts, United States) equipped with a PFA filter folder was placed between the valve and bubblers to prevent water droplets from entering the pump. In addition, it was equipped with a gas flow multi-meter (Model 5210, TSI Incorporated, Minnesota, United States) that can measure pressure, temperature, flow rate, and integrated flow rate, as well as a pump (DAP-12S, ULVAC, Kanagawa, Japan) for atmospheric suction.

### Laboratory experiment

**NOx collection.** To test the NOx collection efficiency, the pump was replaced with a NOx analyser (APNA-370, HORIBA, Kyoto, Japan or Serinus 40, ACOEM ECOTECH, Melbourne, Australia) and a multi-meter was moved in front of the PTFE filter. A three-port valve was installed to bypass the three bubblers. The difference in concentrations between the two instruments was within 1 ppb for NO, $NO_2$, and NOx at atmospheric concentrations ($<$ 90 ppb). Cylinders containing NO or $NO_2$ were diluted in two steps using pure $N_2$ equipped with a mass flow controller (Kofloc, Kyoto, Japan) to prepare approximately 15 and 40 ppb NOx gas. The collection periods were one hour for each sampling and the flow rate was 0.8 L $min^{-1}$. For the sample collection, the 10 mL of recovery solution was placed in three bubblers. For blanks, 5 mL of the solution was placed in three bubblers and collected immediately without sucking air and the blank was collected every 2–3 sample collection. Room temperature was maintained at 22–30°C. Prior to entering the collection system, some of the diluted NO or $NO_2$ gas was vented outside to reduce the pressure and adjust the sampling pressure to atmospheric pressure (1000–1020 hPa). After pumping, all recovered solutions were transferred to a 60 mL amber plastic bottle. The collection efficiency was calculated from the reduction in NO and $NO_2$ concentrations measured by the NOx analyser with and without passing through the three bubblers.

For the $H_2O_2$/NaOH recovery solution, after transferring the recovery solution to a 60 mL amber plastic bottle, $H_2O_2$ was removed from the recovery solution by adding manganese oxide ($MnO_2$) (Special grade, FUJIFILM Wako Pure Chemical Corp., Osaka, Japan) to stop the reaction [24]. It is necessary to slightly loosen the lid of the plastic bottle or periodically open it because oxygen is generated. Hydrogen chloride (HCl) was added to neutralize the recovery solution.

For the $KMnO_4$/NaOH recovery solution, we followed the method reported by Fibiger et al. [14]. After transferring the recovery solution to a 60 mL amber plastic bottle for storage, the solution was transferred to a well-washed 500 mL glass beaker. Then, a total of 10 mL of 35 wt% $H_2O_2$ was added to reduce $KMnO_4$. When incorporating $H_2O_2$, exercise caution to use a 500 mL glass beaker with a wide mouth instead of the designated 60 mL amber plastic bottle. Failure to do so may result in an abrupt release of the solution. After stopping the reaction, the clear solution and brown $MnO_2$ were transferred to a 50 mL centrifuge tube. Then, the solutions were added HCl to neutralize and centrifuged at 4800 rpm for 15 minutes. After centrifugation, the supernatant was collected. $MnO_2$ adhering to the beaker can be removed by washing with HCl. Similarly, the sampling system also deposits $MnO_2$ and clogs the frit, requiring HCl to be added to remove the $MnO_2$ after several samples.

**Ammonia impact assessment.** To test the effect of ammonium ion contamination on the $\delta^{15}N$ values of NOx, 100 μM of $NH_4Cl$ (special grade, FUJIFILM Wako Pure Chemical Corp.,

Osaka, Japan) in water was added to the prepared recovery solutions. After seven days, $MnO_2$ or $H_2O_2$ was added to the recovery solution to stop the reaction. Continuous flow analysis (CFA) (QuAAtro-2HR, BL TEC K.K., Tokyo, Japan) was used to compare the effect of ammonia mixing by measuring the concentrations of nitrate and nitrite ions in the samples with and without added ammonia.

## Field measurement

Ambient $NO_x$ samples were collected at Tsukuba in the west office of AIST (Tsukuba site), Ibaraki, Japan (36.05˚N, 140.12˚E, 12 m above ground level), and Tokai University in Shibuya (Yoyogi site), Tokyo, Japan (35.66˚ N, 139.68˚ E, 52 m above ground level) from January to February 2023 on weekdays (S1 Fig). The Tsukuba site does not require permission as it is the author's affiliated institution. Access to the Yoyogi site was granted by Tokai University. We confirmed that the field studies did not involve endangered or protected species. The flow rate ranges from 0.5 to 0.8 L min$^{-1}$. After confirming that the $NO_x$ concentration did not change, a PFA or Dekabon tube was used to connect the air inlet to the sampling system. It was confirmed at Tsukuba that more than 95% of the $NO_x$ had been collected by branching the collection system and measuring the $NO_x$ concentrations before and after passing the collection system. The $NO_2^-$ and $NO_3^-$ blanks were about 0.5 μM in the Tsukuba and the Yoyogi sites.

**Isotopic analysis.** Ten nmol of nitrate and/or nitrate ions in the obtained recovery solution were measured using the denitrifier method [25, 26]. If a small amount of $MnO_2$ (about 0.2 g) was added stopping the reaction of the $H_2O_2$/NaOH recovery solution, the isotope ratio was not affected even if $MnO_2$ dissolved in about 20 mL of solution was directly added to the vial. The automated injection line was modified from Hattori et al. [27] and a schematic diagram of the injection system is depicted in S2 Fig. Briefly, $NO_2^-$ and $NO_3^-$ were converted to nitrous oxide ($N_2O$) by a strain of denitrifying bacteria, *Pseudomonas aureofaciens*, which has no $N_2O$ reductase. The $N_2O$ produced was then separated from other chemical species using chemical traps and a column equipped with a gas chromatograph (HP-plot Q; Agilent Technologies, Inc., California, United States) and was measured using an isotope-ratio mass spectrometer (IRMS) (MAT252; Thermo Fisher Scientific Inc., Massachusetts, United States) with industrial helium (99.99% purity) as carrier gas. Internationally recognized $NO_3^-$ reference standards USGS 32, 34, 35 and their mixtures were measured alongside the samples and used to correct the resulting mass 45/44 and 46/44 ratios to obtain the final $\delta^{15}N$ and $\delta^{18}O$, respectively. The 1σ analytical uncertainty of $\delta^{15}N$ and $\delta^{18}O$ values were ± 0.5 and ± 0.8 ‰, respectively. The low purity of helium affects the measurement precision, but industrial helium was used because ultrapure helium is expensive and not easily available. Although we confirmed with IRMS that the baseline for m/z 44, 45, and 46 did not increase when changing from ultrahigh purity helium to industrial-purity helium, the standard deviation (1σ) of the standard $\delta^{15}N$ value deteriorated by 0.2‰. Impurities contained in helium were likely concentrated during the $N_2O$ purge and trap process. Although ultrahigh-purity helium was more suitable for isotope measurement, depending on the molecule, industrial-purity helium was deemed suitable for measurements with little deviation from the blank. Adding $MnO_2$ to $NO_3^-$ reference standards did not change the precision and accuracy of $\delta^{15}N$ values.

**Definition.** Stable isotopic compositions are typically reported as:

$$\delta X_{sample} = \frac{R_{sample}}{R_{standard}} - 1, \tag{7}$$

where $X$ denotes $^{18}O$, and $^{15}N$, and $R$ represents the ratios of $^{18}O/^{16}O$, and $^{15}N/^{14}N$ in either the sample or standard material. The δ values are often quoted using per mil (δ) notation. The

$\delta^{15}N$ value was relative to atmospheric $N_2$ (air), whereas the $\delta^{18}O$ value is relative to Vienna Standard Mean Ocean Water (VSMOW). After analysis of the sample recovery solution and blank, the final sample isotopes were calculated using the mass balance:

$$\delta X_{sample} = (\delta X_{total}[NO_3^-]_{total} - \delta X_{blank}[NO_3^-]_{blank})/([NO_3^-]_{total} - [NO_3^-]_{blank}) \qquad (8)$$

where $\delta X_{total}$ and $\delta X_{blank}$ were determined by IRMS with the sample and blank measurement, respectively. $[NO_3^-]_{total}$ and $[NO_3^-]_{blank}$ were determined by CFA or IRMS with the sample and blank measurement, respectively. For the atmospheric $NO_x$ sample, to ensure precise and accurate measurement of the $\delta^{15}N$ values, we considered the $\Delta^{17}O$ ($\Delta^{17}O = \delta^{17}O - 0.52 \times \delta^{18}O$) [28] of the analyte $N_2O$.

## Results and discussion

### Collection efficiency

Nitrogen isotope exchange between NO and $NO_2$ has been suggested to influence N stable isotope compositions. For accurate $NO_x$ isotopic composition measurements, nitrogen isotope analysis of atmospheric $NO_x$ requires the collection of both NO and $NO_2$ with high collection efficiency. Furthermore, because NO and $NO_2$ have different physical properties, a differential assessment of NO and $NO_2$ collection efficiencies is required. The $NO_x$ collection efficiencies are listed in Table 1. After the experiment, like Fibiger et al. [14], the volume of the solution decreased by a few mL, indicating droplet dispersal. To prevent loss of nitrate due to droplet scattering, three bubblers were used, although the collection rate did not differ considerably when two bubblers were used. Although the effect of this decrease in water content on the isotopic composition of nitrate is difficult to estimate, the concentration of nitrate in the third bubbler was the same as that in the blank. Droplet dispersal is mainly affected by the third-stage bubbler, but since the third stage has a low $NO_x$ concentration, the effect of droplet dispersal on the isotopic composition of nitrate was deemed to be negligible.

No significant difference was found when comparing the collection efficiency of both recovery solutions at $NO_x$ concentrations of 15 and 40 ppb (Table 1). The overall averaged $H_2O_2/NaOH$ recovery solution collected over 90% of NO and over 95% of $NO_2$. Over 90% of NO and $NO_2$ collection efficiency using $H_2O_2/NaOH$ recovery solution was also reported by Ohta et al. [18], when the concentration of the recovery solution was over 0.6% $H_2O_2$ and 0.24 M NaOH. Note that while air is flowing, $CO_2$ reacts with NaOH in solution, lowing the pH

**Table 1. Collection efficiency of NO and $NO_2$ by $H_2O_2$/ NaOH and $KMnO_4$/NaOH recovery solution.**

| | Recovery solution (10 mL solution in three bubblers) | |
|---|---|---|
| Gas | $H_2O_2/NaOH$[a] | $KMnO_4/NaOH$[b] |
| *15–20 ppb $NO_x$* | | |
| NO | 97 ± 4% (n = 5) | 85 ± 3% (n = 4) |
| $NO_2$ | 99 ± 3% (n = 5) | 91 ± 3% (n = 4) |
| *30–40 ppb $NO_x$* | | |
| NO | 94 ± 2% (n = 8) | 83 ± 4% (n = 5) |
| $NO_2$ | 97 ± 3% (n = 8) | 91 ± 2% (n = 5) |

[a]$H_2O_2/NaOH$ recovery solution was prepared as a mixture of 3 wt% $H_2O_2$ and 0.5 M NaOH. [b]$KMnO_4/NaOH$ was prepared by mixtures of 0.25 M $KMnO_4$ and 0.5 M NaOH. The average and standard deviation (1σ) of collection efficiency was calculated from the reduction in NO and $NO_2$ concentrations measured by the $NO_x$ analyzer with and without passing through the three gas bubblers.

[21, 22]. Thus, the concentration of NaOH should be greater than 0.24 M. More than 95% of NO$_x$ can be collected by using H$_2$O$_2$/NaOH recovery solution since NO$_2$/NO$_x$ in the atmosphere mostly exceeds 50%. The high collection efficiency of the H$_2$O$_2$/NaOH recovery solution probably made isotopic fractionation negligible during sampling. However, it is necessary to consider the differences in isotopic composition due to the differences in the collection efficiencies of NO and NO$_2$. The reported $\delta^{15}$N(NO$_2$) values ranges from −22 to 5‰ [10–12, 17]. Since fractionation factors for $^{15}$N substitution between NO and NO$_2$ ranged from 1.040 at 278 K to 1.034 at 310 K [29], when the $\delta^{15}$N(NO$_2$) values are from −22 to 0.4 ‰, the expected $\delta^{15}$N(NO) values is higher by approximately 40‰ at 278 K. The apparent $\delta^{15}$N(NO$_x$) value can be increased by a maximum of 0.74 ‰ compared to the true value by the difference in the collection efficiency of NO and NO$_2$ when the mole fraction of NO$_2$ to NO$_x$ is 0.49.

The KMnO$_4$/NaOH recovery solution captured approximately 83–85% of NO and 91% of NO$_2$. Given that the mole fraction of NO$_2$ is much larger than NO, the collection efficiency of NO$_x$ in KMnO$_4$/NaOH recovery solution was in good agreement with the reported value (92 ± 10) % [14, 15]. In this study, both recovery solutions can collect NO$_x$ with high efficiency, and H$_2$O$_2$/NaOH has a higher absorption efficiency than KMnO$_4$/NaOH under the same conditions. However, we have not evaluated the extent to which recovery solution affects isotope ratio fractionation.

## Nitrite and nitrate concentration in the recovery solution

Blank reduction is important in NO$_x$ isotope measurements. We compared the NO$_2^-$ and NO$_3^-$ blanks in H$_2$O$_2$/NaOH with those in the KMnO$_4$/NaOH recovery solution. The NO$_2^-$ and NO$_3^-$ concentrations of the blanks were measured using CFA, and the results are listed in Table 2. The blanks of the H$_2$O$_2$/NaOH and KMnO$_4$/NaOH recovery solution were approximately 0.5 μM and 2.8 μM, respectively. The blank concentrations of NO$_2^-$ and NO$_3^-$ in the H$_2$O$_2$/NaOH recovery solution were clearly lower than the values of KMnO$_4$/NaOH, indicating that the H$_2$O$_2$/NaOH recovery solution is superior in blank suppression. Fibiger et al. [14] also tried NO$_x$ collection using an H$_2$O$_2$/NaOH recovery solution and found a high nitrate blank (approximately 25 μM). However, such high-level blanks in the H$_2$O$_2$/NaOH recovery solution were not observed when using the reagents or experimental scheme presented in this study.

In addition, 120 μM NH$_4$Cl was added to KMnO$_4$/NaOH and H$_2$O$_2$/NaOH recovery solutions and allowed to stand for one week to investigate the effect of ammonium ions. Table 2 presents the results of the study. Total NO$_2^-$ and NO$_3^-$ concentration increased in the KMnO$_4$/NaOH recovery solution but not in the H$_2$O$_2$/NaOH recovery solution. As described by Fibiger et al. [14] and Wojtal et al. [15], the KMnO$_4$/NaOH recovery solution slightly reacted with ammonia ions after seven days, but the H$_2$O$_2$/NaOH recovery solution did not react with ammonia ions. However, since neither of the recovery solutions reacted by even 1% of the amount of ammonia added, it is hypothesized that a negligible reaction occurred with ammonia. From these results, it is evident that the H$_2$O$_2$/NaOH recovery solution is superior to the KMnO$_4$/NaOH recovery solution in terms of NO$_x$ collection. Subsequent experiments were performed using the H$_2$O$_2$/NaOH recovery solution only and had a high collection rate and suppressed blanks. Under the same experimental conditions, the H$_2$O$_2$/NaOH recovery solution outperforms the KMnO$_4$/NaOH recovery solution. A further advantage over previous methods is that neutralization can be performed in one vessel, and no centrifugation is required, resulting in a reduced risk of sample loss and contamination.

**Table 2. Nitrite and nitrate concentrations of the blank and seven days later after adding ammonium ion in recovery solution.**

| Recovery solutions | | NO$_2^-$ + NO$_3^-$ (μM)[a] | Averaged NO$_2^-$ + NO$_3^-$ (μM) |
|---|---|---|---|
| **Blank** | | | |
| | H$_2$O$_2$/NaOH[b]_1 | 0.4 | 0.5 ± 0.2 |
| | H$_2$O$_2$/NaOH_2 | 0.8 | |
| | H$_2$O$_2$/NaOH_3 | 0.4 | |
| | H$_2$O$_2$/NaOH_4 | 0.2 | |
| | H$_2$O$_2$/NaOH_5 | 0.5 | |
| | H$_2$O$_2$/NaOH_6 | 0.5 | |
| | H$_2$O$_2$/NaOH_7 | 0.5 | |
| | KMnO$_4$/NaOH_1 | 1.1 | 1.5 ± 0.6 |
| | KMnO$_4$/NaOH_2 | 1.0 | |
| | KMnO$_4$/NaOH_3 | 1.7 | |
| | KMnO$_4$/NaOH_4 | 1.4 | |
| | KMnO$_4$/NaOH_5 | 2.3 | |
| **After adding 100 μM ammonium ion[d]** | | | |
| | H$_2$O$_2$/NaOH_1 | 0.7 | 0.5 ± 0.2 |
| | H$_2$O$_2$/NaOH_2 | 0.6 | |
| | H$_2$O$_2$/NaOH_3 | 0.4 | |
| | H$_2$O$_2$/NaOH_4 | 0.6 | |
| | H$_2$O$_2$/NaOH_5 | 0.3 | |
| | H$_2$O$_2$/NaOH_6 | 0.4 | |
| | H$_2$O$_2$/NaOH_7 | 0.4 | |
| | KMnO$_4$/NaOH_1 | 3.2 | 2.8 ± 0.9 |
| | KMnO$_4$/NaOH_2 | 2.4 | |
| | KMnO$_4$/NaOH_3 | 2.1 | |
| | KMnO$_4$/NaOH_4 | 2.7 | |
| | KMnO$_4$/NaOH_5 | 1.9 | |

[a]The NO$_2^-$ and NO$_3^-$ concentrations in recovery solution were measured by the CFA method. [b]H$_2$O$_2$/NaOH recovery solution was a prepared mixture of 3 wt% H$_2$O$_2$ and 0.5 M NaOH. [c]KMnO$_4$/NaOH recovery solution was a prepared mixture of 0.25 M KMnO$_4$ and 0.5 M NaOH.

[d]The reaction was stopped seven days after the addition of ammonium ions, and NO$_2^-$ and NO$_3^-$ concentrations in the recovery solution were measured.

## Nitrogen isotope measurement for cylinder NO$_x$

The δ$^{15}$N values of samples A and B recovered with the H$_2$O$_2$/NaOH solution were measured, as shown in Table 3. The repeatability of δ$^{15}$N values for samples A and B were (0.7 ± 0.5) ‰ ($n$ = 5) and (−18.0 ± 0.8) ‰ ($n$ = 5) for 15–20 ppb of NO$_x$, respectively. The repeatability of δ$^{15}$N values for samples A and B were (0.8 ± 0.5) ‰ ($n$ = 8) and (−17.7 ± 0.7) ‰ ($n$ = 8) for 30–40 ppb of NO$_x$, respectively. The larger uncertainty of the δ$^{15}$N values for sample B compared to those of sample A is thought to be due to slight contamination from ambient air.

## Quantification of the influence of Δ$^{17}$O on δ$^{15}$N values

The δ$^{15}$N value of N$_2$O measured by IRMS is calculated assuming Δ$^{17}$O (= δ$^{17}$O − 0.52 δ$^{18}$O) is 0‰. However, since the Δ$^{17}$O value of N$_2$O may apparently increase the δ$^{15}$N value, a correction was performed taking the Δ$^{17}$O value into account. In this study, the Δ$^{17}$O value of NO$_3^-$ was not measured. On the other hand, the maximum δ$^{18}$O value of N$_2$O converted from NO$_x$

**Table 3. Reproducibility of $\delta^{15}$N value for NO and $NO_2$ in the cylinders using $H_2O_2$/NaOH recovery solution.**

| | Number of experiments | NOx cylinder | Averaged [NOx](ppb) | $\delta^{15}$N (‰) | Averaged Blank/ total N[c] |
|---|---|---|---|---|---|
| 15–20 ppb NO | 5 | A[a] | NOx:15 | 0.7 ± 0.5 | 0.13 |
| | | | (NO:14, NO2:0) | | |
| 30–40 ppb NO | 8 | A | NOx:32 | 0.8 ± 0.5 | 0.12 |
| | | | (NO:31, NO2:1) | | |
| 15–20 ppb NO2 | 5 | B[b] | NOx:16 | −18.0 ± 0.8 | 0.06 |
| | | | (NO:2, NO2:14) | | |
| 30–40 ppb NO2 | 8 | B | NOx:37 | −17.7±0.7 | 0.09 |
| | | | (NO:7, NO2:30) | | |

[a]Sample A was 91 ppm NO balanced with $N_2$ in a cylinder.

[b]Sample B was 5 ppm $NO_2$ balanced with $N_2$ in a cylinder. These high concentrations of NO or $NO_2$ gases were diluted by pure $N_2$ and the diluted concentration was approximately 40 ppb for NOx. The collection periods were one hour for each sampling and flow rates were 0.8 L min$^{-1}$.

[c]The blank/total N values were calculated from the sample and blank peak areas using MAT252.

was 40‰ (containing laboratory and field experiments). Note that the $\delta^{18}$O value of $N_2$O does not directly reflect the $\delta^{18}$O value of NOx because NOx obtains oxygen derived from water or $H_2O_2$ during oxidation in the recovery solution. The $\Delta^{17}$O values are generated only by mass transfer of O atoms from ozone to products during oxidation reactions [30]. In Eqs 1–6, the $\Delta^{17}$O of O supplied during the process of NO and $NO_2$ oxidizing $NO_3$ is 0‰. Considering that NO and $NO_2$ each receive two oxygen atoms, the $\Delta^{17}$O values become 2/3 or less (Eqs 1–6). Further, it is assumed that the relationship between the $\Delta^{17}$O and $\delta^{18}$O values of $NO_2$ is 0.36:1 (estimated by a straight line passing through the origin based on the values reported by Albertin et al. [12]). At this time, the maximum $\Delta^{17}$O value of $N_2$O was 9.5‰ when the $\delta^{18}$O value was 40‰. The effect of this $\Delta^{17}$O value on the $\delta^{15}$N value was estimated using USGS34 and USGS35 with known $\Delta^{17}$O values. Based on the result of USGS35 measurements, when the $\Delta^{17}$O value is 21.56‰ [28], the apparent $\delta^{15}$N value would increase by 1.2‰. This result showed good agreement with the description by Yu and Elliott [16]. Assuming a linear relationship between $\Delta^{17}$O and $\delta^{15}$N values of USGS34 ($\Delta^{17}$O value: −0.3‰ [31]) and USGS35, the $\Delta^{17}$O value of 9.5‰ will increase the $\delta^{15}$N value by 0.5‰ at maximum. The overall measured 1σ uncertainty of $\delta^{15}$N(NOx) was ± 1.2‰ by combining the difference in the absorption efficiency of NO and $NO_2$, the repeatability, and the consideration of $\Delta^{17}$O value.

## Limitations of NOx collection

We also tested the limitations of this developed method. Possible factors that reduce the yield of NOx include a decrease in oxidant concentration due to reaction with NOx and other gases, and a decrease in pH due to reaction with $CO_2$. The reaction between NOx and the oxidant agent is not rate-limiting as the input $H_2O_2$ concentration is sufficiently high compared to the NOx concentration. In fact, we tried flowing 40 ppb of NO and $NO_2$ for over 12 h each, but the collection efficiency did not fall below 90%. On the other hand, when the time required for the collection rate to drop below 90% was measured for continuous collection of approximately 15 ppb of NOx in air at an average of 0.6 L min$^{-1}$, the collection rate dropped sharply at 14 h (S3 Fig). $CO_2$ dissolves in the form of $CO_3^{2-}$ at high pH as follows:

$$CO_2 + 2OH^- \rightarrow CO_3^{2-} + H_2O. \qquad (9)$$

Since it is difficult to calculate the dissolution rate of $CO_2$ in this study, we assumed that all $CO_2$ dissolves in solution. In addition, although $OH^-$ ions are used or provided in the

decomposition of $H_2O_2$, this was not accounted for. We set the flow rate at 0.6 L min$^{-1}$ and the $CO_2$ concentration at 400 ppm. Given that the NO collection efficiency decreases under pH 11 [21] and that the two containers were sufficient to collect $NO_x$, the combined NaOH from the two containers would be neutralized in about 8 h. The actual capacity was longer than 8 h, as it is unknown whether all the $CO_2$ will be absorbed and whether the first bubbler can continue to collect $CO_2$ even after neutralization. When collecting $NO_x$ from air, we recommend up to 8 and 6 h for the collection time, at a flow rate of 0.6 and 0.8 L min$^{-1}$, respectively. If used in an environment with high $CO_2$ concentration, the flow rate must be reduced, or the number of bubblers must be increased.

## Comparison with the previous $NO_x$ isotope measurement method

The advantage of this method is that both NO and $NO_2$ can be collected with a low blank of $NO_3^-$; thus, the $\delta^{15}N$ values of atmospheric $NO_x$ can be directly estimated compared to the methods that can only collect $NO_2$. An offline method for converting high concentrations (over 100 ppm) of $NO_x$ to $NO_3^-$ using $H_2O_2$/NaOH recovery solution was used by Heaton [32]. However, the concentration of $H_2O_2$ and NaOH in the recovery solution is unknown. This study is the first to investigate the $NO_x$ collection efficiency, the degree of blanking, and the influence of ammonium ions being quantified using $H_2O_2$/NaOH recovery solution. Another offline wet method for collecting high concentrations of $NO_x$ is the use of $H_2SO_4$/$H_2O_2$. However, Chin et al. [33] showed that the 6 wt% $H_2O_2$ in low pH (2 to 4) converted less than 5% of NO in the flue gas, and Ohta et al. [18] showed that the $NO_2$ collection efficiency using $H_2O_2$/NaOH recovery solution can be degraded at NaOH concentrations below 0.24 M, suggesting that probably the $H_2O_2$ in basic solution is necessary for high collection efficiency of $NO_x$ online. Further advantages of this method are its portability, simplicity, and the ability to collect the required amount of sample to analyze the nitrogen isotopic composition of ambient $NO_x$ in a short period of time.

## Nitrogen isotope measurement for atmospheric $NO_x$

The observed $\delta^{15}N$ values and $NO_x$ concentrations for atmospheric $NO_x$ collected at the Tsukuba and Yoyogi sites are shown in Fig 2 and S2 Table. The average $NO_x$ concentrations during the sampling period were 6 and 18 ppb at Tsukuba and Yoyogi, respectively. The maximum $NO_x$ concentrations at Tsukuba and Yoyogi were 45 and 143 ppb, respectively. The diurnal variation in the $NO_x$ concentration on weekdays during the sampling period clearly showed two peaks corresponding to traffic rush hours (S4 and S5 Figs).

The $\delta^{15}N(NO_x)$ value ranged from −3.1 to −0.5 ‰ at the Tsukuba site and from −5.6 to −0.5 ‰ at the Yoyogi site. The averaged $\delta^{15}N(NO_x)$ value and standard deviation (1σ) in the Yoyogi site was (−2.7 ± 1.8) ‰ and in the Tsukuba site was (−1.7 ± 0.9) ‰ during the sampling period, and no significant difference between the two sites was observed. No significant correlation was found between the $NO_x$ concentrations or $1/[NO_x]$ and $\delta^{15}N(NO_x)$ values. Additionally, it also did not correlate with the ratio of $NO_2$ to $NO_x$ observable when only the $\delta^{15}N(NO_2)$ was measured because we collected both NO and $NO_2$.

Walters et al. [34] showed the mass-weighted $\delta^{15}N(NO_x)$ values emitted from vehicles have the following relationship with vehicle runtime:

$$\delta^{15}N(NO_x) = -12.35 + 3.02\ln(t + 0.455) \qquad (10)$$

where $\delta^{15}N(NO_x)$ represents the mass-weighted $\delta^{15}N(NO_x)$ values emitted from vehicles and $t$ is the vehicle run time (min). Because the average distance of one car in Japan is approximately 20 km (Ministry of Land, Infrastructure, Transport and Tourism website, https://www.e-stat.

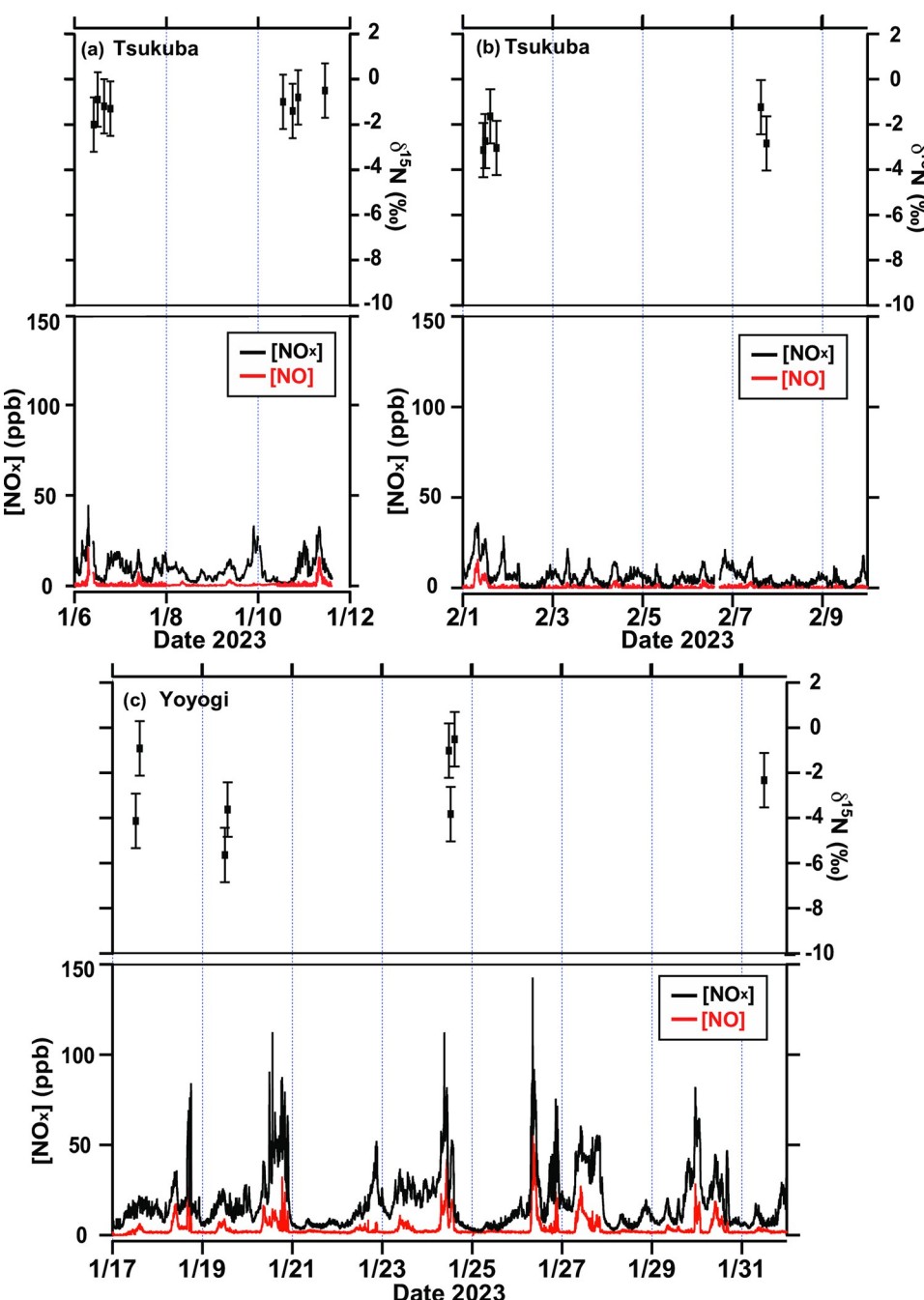

**Fig 2.** NO (red) and NO$_x$ (black) concentrations and $\delta^{15}$N(NO$_x$) values at the Tsukuba (a, b) and Yoyogi site (c). Error bars indicate 1σ uncertainty of $\delta^{15}$N(NO$_x$) ± 1.2‰.

go.jp/stat-search/files?stat_infid=000032211818, The first summary table by fuel/vehicle type last access: 28 March 2023 [35]), the driving time is expected to be 10−30 mins each. The predicted mass-weighted $\delta^{15}$N(NO$_x$) values emitted from vehicles (−5.3 to −2.0 ‰; S1 Table) based on Eq 10 were matched with the $\delta^{15}$N(NO$_x$) values in the Tsukuba and the Yoyogi sites. Therefore, if the main source of NO$_x$ is vehicle exhaust, then the $\delta^{15}$N(NO$_x$) values and diurnal variations in NO$_x$ concentrations can be explained. Biomass burning (−7 to 12 ‰ [36];

S1 Table) is also a candidate for the NO$_x$ sources of observed $\delta^{15}$N(NO$_x$) values in the Tsukuba and the Yoyogi sites. However, because the NO$_x$ emitted from biomass burning is temporary, it is unlikely to be the main source of NO$_x$ in urban areas. Additionally, biomass burning did not show diurnal variation in NO$_x$ concentrations, as shown in S4 and S5 Figs. This indicates that the main NO$_x$ source collected at the Tsukuba and Yoyogi sites was vehicle exhaust during the sampling period. However, the sampling period is limited and not all variations in $\delta^{15}$N (NO$_x$) value can be explained. Future investigations are needed to understand NO$_x$ dynamics by measuring the $\delta^{15}$N(NO$_x$) value of NO$_x$ sources to enrich the database in the surrounding environment and through long-term observations with higher time resolution.

## Conclusion

We developed a portable new method to collect NO$_x$ for nitrogen isotopic measurement by mixing 3 wt% H$_2$O$_2$ and 0.5 M NaOH solution with a precision (1$\sigma$ uncertainty) of ± 1.2 ‰. The method using the developed H$_2$O$_2$/NaOH recovery solution has high NO$_x$ collection efficiency, a relatively simpler measurement procedure, small blanks, and a negligible impact of ammonium contamination.

The $\delta^{15}$N(NO$_x$) values were observed in two sites in Japan. The averaged $\delta^{15}$N(NO$_x$) value and standard deviation (1$\sigma$) in the Yoyogi site was (−2.7 ± 1.8) ‰ and in the Tsukuba site was (−1.7 ± 0.9) ‰ during the sampling period. The main NO$_x$ source appears to be the vehicle exhaust in the two sites. However, the sampling period is limited and the not all variations in $\delta^{15}$N(NO$_x$) value can be explained. Future investigations are needed to understand NO$_x$ dynamics by measuring the $\delta^{15}$N(NO$_x$) value of NO$_x$ sources to enrich the database in the surrounding environment and through long-term observations with higher time resolution. In addition, by combining the concentration and $\delta^{15}$N value of ammonia, organic nitrogen, and nitrate in aerosols, among others, the understanding of the nitrogen cycle, including NO$_x$ will be deepened.

## Supporting information

**S1 Fig. Aerial view of the sampling sites.**
(PDF)

**S2 Fig. Schematic diagram of the system for measuring nitrogen isotope ratios within N$_2$O.**
(PDF)

**S3 Fig. Limitation of NO$_x$ collection.** A NO$_x$ analyzer was connected behind the bubbler and when approximately 15 ppb of NO$_x$ in the air was continuously captured at an average rate of 0.6 L min$^{-1}$, the time required for the collection efficiency to drop below 90% was measured. We set the bubbler at time 0 min.
(PDF)

**S4 Fig. Box-and-whisker plots of diurnal variation of NO$_x$ concentrations during sampling periods (1/4-6, 10–11, 31, 2/1-3, 7–10) at the Tsukuba site.**
(PDF)

**S5 Fig. Box-and-whisker plots of diurnal variation of NO$_x$ concentrations during sampling periods (1/17-20, 24–27, 30–31) at the Yoyogi site.**
(PDF)

**S1 Table. Overview of nitrogen isotopic composition ($\delta^{15}$N) for nitrogen oxides (NO$_x$) in the atmosphere.**
(XLSX)

**S2 Table. The $\delta^{15}$N(NO$_x$) and NO$_x$ concentration of samples collected in ambient urban air at Tsukuba and Yoyogi site from January to February 2023.**
(XLSX)

## Acknowledgments

We thank Tokai University for providing the observation sites. We thank Taizo Sano at the National Institute of Advanced Industrial Science and Technology (AIST), Japan, for allowing the use of the device. We also thank Shohei Hattori at the Tokyo Institute of Technology, Japan (current address: International Center for Isotope Effects Research Nanjing University, China) for the help with the isotopic composition measurement method for nitrate using the denitrifier method.

## Author Contributions

**Conceptualization:** Kazuki Kamezaki.

**Data curation:** Kazuki Kamezaki, Takahisa Maeda, Shigeyuki Ishidoya, Ayumi Tsukasaki.

**Formal analysis:** Kazuki Kamezaki, Takahisa Maeda, Shigeyuki Ishidoya, Ayumi Tsukasaki.

**Funding acquisition:** Kazuki Kamezaki, Shigeyuki Ishidoya.

**Investigation:** Kazuki Kamezaki, Takahisa Maeda, Shigeyuki Ishidoya.

**Methodology:** Kazuki Kamezaki, Shigeyuki Ishidoya.

**Project administration:** Kazuki Kamezaki.

**Resources:** Kazuki Kamezaki.

**Software:** Takahisa Maeda.

**Supervision:** Shigeyuki Ishidoya.

**Validation:** Kazuki Kamezaki, Shigeyuki Ishidoya.

**Visualization:** Kazuki Kamezaki.

**Writing – original draft:** Kazuki Kamezaki.

**Writing – review & editing:** Kazuki Kamezaki, Takahisa Maeda, Shigeyuki Ishidoya, Ayumi Tsukasaki, Shohei Murayama, Naoki Kaneyasu.

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
