## [Decision Letter · Decision Letter 0]

5 Nov 2023

PONE-D-23-29515Low blank sampling method for measurement of the nitrogen isotopic composition of atmospheric NOxPLOS ONE

Dear Dr. Kamezaki,

Thank you for submitting your manuscript to PLOS ONE. After careful consideration, we feel that it has merit but does not fully meet PLOS ONE’s publication criteria as it currently stands. Therefore, we invite you to submit a revised version of the manuscript that addresses the points raised during the review process.

We look forward to receiving your revised manuscript.

Kind regards,

Eva Elisabeth Stüeken, Ph.D.

Academic Editor

PLOS ONE

Journal Requirements:

"This study was supported by a Grant from the Steel Foundation for Environmental Protection Technology (FY2021–2022) (K. K.) and Grants-in-Aid for Scientific Research 22H05006 (S.I.) and 22H03739 (K.K. and S.I.) from the Ministry of Education, Culture, Sports, Science, and Technology (MEXT), Japan. We thank Tokai University for providing the observation sites. We thank Taizo Sano at the National Institute of Advanced Industrial Science and Technology (AIST), Japan, for allowing us to use this device. We also thank Shohei Hattori at the Tokyo Institute of Technology, Japan (current address: International Center for Isotope Effects Research Nanjing University, China) for teaching us the isotopic composition measurement method for nitrate using the denitrifier method."

"This study was supported by a Grant from the Steel Foundation for Environmental Protection Technology (FY2021–2022) (K. K.) and Grants-in-Aid for Scientific Research 22H05006 (S.I.) and 22H03739 (K.K. and S.I.) from the Ministry of Education, Culture, Sports, Science, and Technology (MEXT), Japan. The funders had no role in study design, data collection and analysis, decision to publish, or preparation of the manuscript"

5. We note that Figure S1 in your submission contain map/satellite images which may be copyrighted. All PLOS content is published under the Creative Commons Attribution License (CC BY 4.0), which means that the manuscript, images, and Supporting Information files will be freely available online, and any third party is permitted to access, download, copy, distribute, and use these materials in any way, even commercially, with proper attribution. For these reasons, we cannot publish previously copyrighted maps or satellite images created using proprietary data, such as Google software (Google Maps, Street View, and Earth). For more information, see our copyright guidelines: http://journals.plos.org/plosone/s/licenses-and-copyright.

A. You may seek permission from the original copyright holder of Figure S1 to publish the content specifically under the CC BY 4.0 license.  

B. If you are unable to obtain permission from the original copyright holder to publish these figures under the CC BY 4.0 license or if the copyright holder’s requirements are incompatible with the CC BY 4.0 license, please either i) remove the figure or ii) supply a replacement figure that complies with the CC BY 4.0 license. Please check copyright information on all replacement figures and update the figure caption with source information. If applicable, please specify in the figure caption text when a figure is similar but not identical to the original image and is therefore for illustrative purposes only.

**Additional Editor Comments:**

AE comments:

Dear Authors,

Thank you for your submission! Your manuscript has now been reviewed by two experts in the field, and they only identified a few minor points that should be addressed. In addition, I list a few points below based on my own reading. If these can be resolved, the paper should be suitable for publication in PlOS ONE.

l. 62: Provide a reference for the rapid H2O2 decomposition at high NaOH. That would be useful for others who want to replicate this method.

l. 66: What is the bubbler and what does sucking mean in this context? Please provide more explanation. If this is explained later, please remove the blank description from here and insert it once the setup has been fully described.

ll. 100-103: Please provide a reference for this method of removing H2O2.

ll. 104-106: This procedure is not clear. Why is the solution first stored in an amber plastic bottle and then in a glass beaker? And how much water is added? Note that there is a superscript ^-1 missing after MOhm*cm. Maybe also add the word ‘to’ after ‘transferred’ in l. 105? In l. 106, maybe reorganise to ‘… and 10ml of 35% H2O2 was added…’

ll. 140-141: How did the low-purity He affect the measurements? This would be useful to know for other researchers.

l. 160: When and where does the reaction between NaOH and CO2 happen?

l. 162: What does the number 13 mean in this line?

l. 176: What does collection without bubblers mean? It is not clear how this was set up. If there are no bubblers, is that not equal to simply transferring the gas from the cylinder to the analyser? But why is that then called ‘uncollected’? This term suggests that it should be the gas that evaded collection. Please clarify.

ll. 176-177: What do the superscripts a and b mean before H2O2 and KMnO4, respectively? Note: I later realized that this section may belong to the table that is shown just above. Please make that clearer. Otherwise, it will be incorrect in the published paper. I recommend submitting all tables with captions and annotations as separate documents.

l. 268: delete ‘the’ before ‘not all variations’

l. 275: change ‘fewer’ to ‘smaller’ and ‘impact on’ to ‘impact of’

Fig. 2: Please label the x-axes and add a legend to the figure to explain what the different lines and symbols mean.

Eva Stüeken

Reviewers' comments:

Reviewer's Responses to Questions

**Comments to the Author**

1. Is the manuscript technically sound, and do the data support the conclusions?

Reviewer #1: Yes

Reviewer #2: Yes

2. Has the statistical analysis been performed appropriately and rigorously? 

Reviewer #1: No

Reviewer #2: N/A

3. Have the authors made all data underlying the findings in their manuscript fully available?

Reviewer #1: Yes

Reviewer #2: Yes

4. Is the manuscript presented in an intelligible fashion and written in standard English?

Reviewer #1: Yes

Reviewer #2: Yes

5. Review Comments to the Author

Reviewer #1: The submitted manuscript by Kazuki Kamezaki et al. reports a method to collect atmospheric NOx and measure 15N for it. This method employs a 3 wt% hydrogen peroxide (H2O2) and 0.5 M sodium hydride (NaOH) solution to collect NOx and found a lower blank and a higher collection efficiency than the other method of KMnO4/NaOH recovery solution. The authors have also used this method to collect atmospheric NOx at the Tsukuba and Yoyogi sites in Japan and measure their 15N natural abundance and concentrations. This method is useful for such a collection and isotope analysis and has a potential to be widely used in the future in the study area. The method is generally clearly described and manuscript is clearly written. I have a few specific concerns.

The collection efficiency was determined with approximately 40 ppb NOx gas. However, at most case, atmospheric NOx concentration is much lower than this concentration. How will be if this method is used to for low atmospheric NOX conditions. I would like to see some discussion for this uncertainty.

The method on how to calculate blank is not clearly described in the manuscript.

What statistical method was used for comparison? For table 1, there is not standard error.

Reviewer #2: Summary: The authors present a method for simultaneous collection of NO and NO2 for nitrogen isotope characterization. Their approach successfully captures total NOx, addressing a significant challenge, and demonstrates a low blank, an improvement over previous collection methods. The method is rigorously characterized through both laboratory and field measurements. The study reveals notable diurnal variability in d15N values, aligning with expected traffic patterns. However, the paper lacks quantification of sample collection capacity and duration, constituting a key limitation. With minor revisions, I recommend this work for publication.

Comments:

Lines 23-26: It would be beneficial to include that NOx is also crucial for nutrient deposition.

Lines 26-27: Consider emphasizing the importance of natural sources of NOx, such as biogenic and lightning sources. Addressing their role in remote atmospheres and utilizing d15N for tracking would yield valuable insights into reactive nitrogen chemistry.

Line 28: Note that global NOx emissions have decreased, but the trend is regionally dependent.

Lines 33-35: I recommend referencing "Collection of Nitrogen Dioxide for Nitrogen and Oxygen Isotope Determination – Laboratory and Environmental Chamber Evaluation" by Blum et al., 2023.

Lines 36-38: Please include a citation for the scrubbing method developed by Yu and Elliott in 2017, titled "Novel Method for Nitrogen Isotopic Analysis of Soil-Emitted Nitric Oxide."

Lines 45-46: Consider mentioning the work by Yu and Elliott here as well for comprehensive context.

Lines 60-61: Clarify the duration for which the 35% H2O2 bottle was utilized. Provide recommendations for storage to maintain its effectiveness.

Lines 65: Specify if the 3% H2O2 was used in this study or in a prior one.

Lines 75-76: Elaborate on whether "NO3- evaporates" refers to NOx breakthrough or evaporation. Quantify the amount of NO3- found in subsequent bubblers to address this breakthrough.

Lines 93-95: The scope of the conduced experiments are limited. Verify if the 40 ppb NOx concentration is representative of real-world atmospheric conditions. Can you quantify the collection capacity of the system for NOx collection? Is there a relationship of collection efficiency with collected amounts? Can you make recommendations for how long and for what amounts of NOx the presented method is valid?

Line 140: Address potential corrections for 17O influence (Δ17O). Given the elevated Δ17O values of NO and NO2, it's possible that some O atoms in the NO3- product are preserved, potentially inflating corrected d15N values.

Line 170: Move the table footnotes describing superscripts "a" and "b" to immediately follow the table for clarity.

Line 185: If KMnO4 is a stronger oxidant than H2O2, speculate on why the solution with H2O2 exhibits higher collection efficiency.

6. PLOS authors have the option to publish the peer review history of their article (what does this mean?). If published, this will include your full peer review and any attached files.

Reviewer #1: No

Reviewer #2: No

---

## [Author Response · Author response to Decision Letter 0]

18 Jan 2024

Respond to Editor comments

Thank you for your submission! Your manuscript has now been reviewed by two experts in the field, and they only identified a few minor points that should be addressed. In addition, I list a few points below based on my own reading. If these can be resolved, the paper should be suitable for publication in PlOS ONE.

Reply: Thank you very much for reviewing our manuscript. 

l. 62: Provide a reference for the rapid H2O2 decomposition at high NaOH. That would be useful for others who want to replicate this method.

Reply: We cited Špalek et al. (1982).

Reference

Špalek O, Balej J, Paseka I. Kinetics of the decomposition of hydrogen peroxide in alkaline solutions. J. Chem. Soc., Faraday Trans. 1982; 78: 2349–2359. https://doi.org/10.1039/F19827802349.

l. 66: What is the bubbler and what does sucking mean in this context? Please provide more explanation. If this is explained later, please remove the blank description from here and insert it once the setup has been fully described.

Reply: We moved the sentence to the “NOx collection” section accordingly.

ll. 100-103: Please provide a reference for this method of removing H2O2.

Reply: We cited Jiang et al. (1991) as an additional reference.

Reference

Jiang SP, Ashton WR, Tseung ACC, An observation of homogeneous and heterogeneous catalysis processes in the decomposition of H2O2 over MnO2 and Mn(OH)2. J. Catal. 1991; 131(1): 88–93, https://doi.org/10.1016/0021-9517(91)90325-X.

ll. 104-106: This procedure is not clear. Why is the solution first stored in an amber plastic bottle and then in a glass beaker? And how much water is added? Note that there is a superscript ^-1 missing after MOhm*cm. Maybe also add the word ‘to’ after ‘transferred’ in l. 105? In l. 106, maybe reorganise to ‘… and 10ml of 35% H2O2 was added…’

Reply: We apologize for the ambiguous sentence. We explained the reason for transferring the solution as follows: “When incorporating H2O2, exercise caution to use a 500 mL glass beaker with a wide mouth instead of the designated 60 mL amber plastic bottle. Failure to do so may result in an abrupt release of the solution.” (line 135). Additionally, since only water was used for washing, we rephrased the text accordingly. Furthermore, we changed “… and 10 ml of 35% H2O2 was added…” to “Then, a total of 10 mL of 35 wt% H2O2 was added to reduce KMnO4.” Based on my research, it seems that “-1” is not necessary after MΩcm, so we left it as it is.

ll. 140-141: How did the low-purity He affect the measurements? This would be useful to know for other researchers.

Reply: We added a few sentences starting at line 173 as follows: “Although we confirmed with IRMS that the baseline for m/z 44, 45, and 46 did not increase when changing from ultrahigh purity helium to industrial-purity helium, the standard deviation (1σ) of the standard δ15N value deteriorated by 0.2‰. Impurities contained in helium were likely concentrated during the N2O purge and trap process. Although ultrahigh-purity helium was more suitable for isotope measurement, depending on the molecule, industrial-purity helium was deemed suitable for measurements with little deviation from the blank.”

l. 160: When and where does the reaction between NaOH and CO2 happen?

Reply: We revised the text on line 211 as follows: “Note that while air is flowing, CO2 reacts with NaOH in solution, lowing the pH.”

l. 162: What does the number 13 mean in this line?

Reply: The number 13 referred to the pH. However, we checked recent research, and it seems that NOx can be captured up to pH 11; therefore, we changed 13 to 11.

l. 176: What does collection without bubblers mean? It is not clear how this was set up. If there are no bubblers, is that not equal to simply transferring the gas from the cylinder to the analyser? But why is that then called ‘uncollected’? This term suggests that it should be the gas that evaded collection. Please clarify.

Reply: As pointed out, “uncollected” was difficult to understand. We changed the sentence in Table 1 as follows: “The average and standard deviation (1σ) of collection efficiency was calculated from the reduction in NO and NO2 concentrations measured by the NOx analyzer with and without passing through the three gas bubblers.”

ll. 176-177: What do the superscripts a and b mean before H2O2 and KMnO4, respectively? Note: I later realized that this section may belong to the table that is shown just above. Please make that clearer. Otherwise, it will be incorrect in the published paper. I recommend submitting all tables with captions and annotations as separate documents.

Reply: Thank you for your suggestion. We changed the order of the sentences in the footnote and submitted all tables with captions and annotations as separate documents.

l. 268: delete ‘the’ before ‘not all variations’

Reply: Corrected accordingly.

l. 275: change ‘fewer’ to ‘smaller’ and ‘impact on’ to ‘impact of’

Reply: Corrected accordingly.

Fig. 2: Please label the x-axes and add a legend to the figure to explain what the different lines and symbols mean.

Reply: Corrected accordingly.

Respond to Reviewer #1

Reviewer #1: The submitted manuscript by Kazuki Kamezaki et al. reports a method to collect atmospheric NOx and measure 15N for it. This method employs a 3 wt% hydrogen peroxide (H2O2) and 0.5 M sodium hydride (NaOH) solution to collect NOx and found a lower blank and a higher collection efficiency than the other method of KMnO4/NaOH recovery solution. The authors have also used this method to collect atmospheric NOx at the Tsukuba and Yoyogi sites in Japan and measure their 15N natural abundance and concentrations. This method is useful for such a collection and isotope analysis and has a potential to be widely used in the future in the study area. The method is generally clearly described and manuscript is clearly written. I have a few specific concerns.

Reply: We sincerely appreciate your constructive comments. 

The collection efficiency was determined with approximately 40 ppb NOx gas. However, at most case, atmospheric NOx concentration is much lower than this concentration. How will be if this method is used to for low atmospheric NOX conditions. I would like to see some discussion for this uncertainty.

Reply: Thank you very much for your comments. We conducted additional experiments with lower concentrations (approximately 15 ppb) of NOx. The results were added in Tables 1 and 3. The collection efficiency at low NOx concentration did not significantly differ relative to previous tests. This method could also be applied for NOx concentrations under 40 ppb. 

The method on how to calculate blank is not clearly described in the manuscript.

Reply: We described the detailed blank correction on line 191.

What statistical method was used for comparison? For table 1, there is not standard error.

Reply: We calculated average and standard deviation (1�) and added the standard deviation in Table 1.

Respond to Reviewer#2 

Reviewer #2: Summary: The authors present a method for simultaneous collection of NO and NO2 for nitrogen isotope characterization. Their approach successfully captures total NOx, addressing a significant challenge, and demonstrates a low blank, an improvement over previous collection methods. The method is rigorously characterized through both laboratory and field measurements. The study reveals notable diurnal variability in d15N values, aligning with expected traffic patterns. However, the paper lacks quantification of sample collection capacity and duration, constituting a key limitation. With minor revisions, I recommend this work for publication.

Reply: Thank you very much for your comments. 

Comments:

Lines 23-26: It would be beneficial to include that NOx is also crucial for nutrient deposition.

Reply: We added a sentence on line 35 as follows: “Besides, NOx deposition can enhance ecosystem productivity through fertilization or decrease it through nutrient imbalances and reduce ecosystem biodiversity through acidification and eutrophication (Galloway et al., 2008).”

Reference

Galloway JN, Townsend AR, Erisman JW, Bekunda M, Cai Z, Freney JR. et al. Transformation of the Nitrogen Cycle: Recent Trends, Questions, and Potential Solutions. Science. 2008;320:889-892. DOI:10.1126/science.1136674.

Lines 26-27: Consider emphasizing the importance of natural sources of NOx, such as biogenic and lightning sources. Addressing their role in remote atmospheres and utilizing d15N for tracking would yield valuable insights into reactive nitrogen chemistry.

Reply: We added biogenic production and lightning as natural sources of NOx on line 38 as follows: “In areas affected by human pollution, fossil-fuel combustion from traffic, residential heating, cooking, industry, and energy sectors are the main sources of NOx. On the other hand, as a natural NOx source, biomass burning, biogenic production, and lightning are also important sources of NOx.”

Line 28: Note that global NOx emissions have decreased, but the trend is regionally dependent.

Reply: We apologize for the inaccurate sentence. We added “Global annual” on line 40 and included McDuffie et al., 2020, to the list of references. 

Reference

McDuffie EE, Smith SJ, O'Rourke P, Tibrewal K, Venkataraman C, Marais EA, et al. A global anthropogenic emission inventory of atmospheric pollutants from sector- and fuel-specific sources (1970–2017): an application of the Community Emissions Data System (CEDS). Earth Syst. Sci. Data, 2020; 12: 3413–3442. https://doi.org/10.5194/essd-12-3413-2020.

Lines 33-35: I recommend referencing "Collection of Nitrogen Dioxide for Nitrogen and Oxygen Isotope Determination – Laboratory and Environmental Chamber Evaluation" by Blum et al., 2023.

Reply: We cited Blum et al. (2023) accordingly.

Reference

Blum DE, Walters WW, Eris G, Takeuchi M, Huey LG, Tanner D, et al. Collection of nitrogen dioxide for nitrogen and oxygen isotope determination laboratory and environmental chamber evaluation. Anal. Chem. 2023; 95: 3371–3378. https://doi.org/10.1021/acs.analchem.2c04672.

Lines 36-38: Please include a citation for the scrubbing method developed by Yu and Elliott in 2017, titled "Novel Method for Nitrogen Isotopic Analysis of Soil-Emitted Nitric Oxide."

Reply: Accordingly, we cited Yu and Elliott (2017).

Reference

Yu Z, Elliott EM. Novel method for nitrogen isotopic analysis of soil-emitted nitric oxide. Environ. Sci. Technol. 2017; 51(11): 6268–6278. https://pubs.acs.org/doi/10.1021/acs.est.7b00592.

Lines 45-46: Consider mentioning the work by Yu and Elliott here as well for comprehensive context.

Reply: We changed the sentence on line 48 as follows: “potassium permanganate (KMnO4) with sodium hydroxide (NaOH) or 20% triethanolamine in water have been used as the recovery solutions for the collection of ambient NOx.”

Lines 60-61: Clarify the duration for which the 35% H2O2 bottle was utilized. Provide recommendations for storage to maintain its effectiveness.

Reply: Thank you for your suggestion. We added a sentence on line 89 as follows: “In our experimental study, it was observed that refrigerated H2O2 remained usable for a period of six months following its purchase. However, after a duration of nine months, the H2O2 failed to generate bubbles even upon NaOH addition.”

Lines 65: Specify if the 3% H2O2 was used in this study or in a prior one.

Reply: In this research, we did not conduct experiments using 3wt% H2O2. We are sorry for the confusion. To avoid misunderstandings, we have removed it from our manuscript.

Lines 75-76: Elaborate on whether "NO3- evaporates" refers to NOx breakthrough or evaporation. Quantify the amount of NO3- found in subsequent bubblers to address this breakthrough.

Reply: Thank you for pointing this out. This evaporation does not mean a NOx breakthrough. Like Fibiger et al. (2014), we also observed the volatilization of 1–2 mL of water from bubblers. Although the effect of this decrease in water content on the isotopic composition of nitrate is difficult to estimate, the concentration of nitrate in the third bubbler was the same as that in the blank. We assumed that the effect of droplet loss on the amount and isotopic composition of nitrate captured was negligible. To describe details, we deleted the sentence and added the following text on line 202: “After the experiment, like Fibiger et al. (2014), the volume of the solution decreased by a few mL, indicating droplet dispersal. To prevent loss of nitric acid due to droplet scattering, three bubblers were used, although the collection rate did not differ considerably when two bubblers were used. Although the effect of this decrease in water content on the isotopic composition of nitrate is difficult to estimate, the concentration of nitrate in the third bubbler was the same as that in the blank. Droplet dispersal is mainly affected by the third stage bubbler, but since the third stage has a low NOx concentration, the effect of droplet dispersal on the isotopic composition of nitrate was deemed to be negligible.”

Lines 93-95: The scope of the conduced experiments are limited. Verify if the 40 ppb NOx concentration is representative of real-world atmospheric conditions. Can you quantify the collection capacity of the system for NOx collection? Is there a relationship of collection efficiency with collected amounts? Can you make recommendations for how long and for what amounts of NOx the presented method is valid?

Reply: Thank you for your comment. We conducted additional experiments at low NOx concentrations, and we added the corresponding data in Tables 1 and 3. The result shows the developed method can be used for low concentrations of NOx.

We also discussed the limitations of this developed method in a new section, which states the limitations of NOx collection. “Possible factors that reduce the yield of NOx include a decrease in oxidant concentration due to reaction with NOx and other gases, and a decrease in pH due to reaction with CO2. The reaction between NOx and the oxidant agent is not rate-limiting as the input H2O2 concentration is sufficiently high compared to the NOx concentration. In fact, we tried flowing 40 ppb of NO and NO2 for over 12 h each, but the collection efficiency did not fall below 90%. On the other hand, a drop in pH can be a limitation. When the time required for the collection rate to drop below 90% was measured for continuous collection of approximately 15 ppb of NOx in air at an average of 0.6 L min−1, the collection rate dropped sharply at 14 h (Fig. S3). In addition, CO2 dissolves in the form of CO32- at high pH as follows:

CO2 + 2OH−　→　CO32− + H2O . (9)

Since it is difficult to calculate the dissolution rate of CO2 in this study, we assumed that all CO2 dissolves in solution. In addition, although OH− ions are used or provided in the decomposition of H2O2, this was not accounted for. We set the flow rate at 0.6 L min−1 and the CO2 concentration at 400 ppm. Given that the NO collection efficiency decreases under pH 11 and that the two containers were sufficient to collect NOx, the combined NaOH from the two containers would be neutralized in about 8 h. The actual capacity was longer than 8 h, as it is unknown whether all the CO2 will be absorbed and whether the first bubbler can continue to collect CO2 even after neutralization. When collecting NOx from the air, we recommend up to 8 and 6 h for the collection time, at a flow rate of 0.6 and 0.8 L min-1, respectively. If used in an environment with high CO2 concentration, the flow rate must be reduced, or the number of bubblers must be increased.”

Fig S3. Limitation of NOx collection. A NOx analyzer was connected behind the bubbler, and when approximately 15 ppb of NOx in the air was continuously captured at an average rate of 0.6 L min-1, the time required for the collection efficiency to drop below 90% was measured. We set the bubbler at time 0 min.

Line 140: Address potential corrections for 17O influence (Δ17O). Given the elevated Δ17O values of NO and NO2, it's possible that some O atoms in the NO3- product are preserved, potentially inflating corrected d15N values.

Reply: Thank you for your suggestion. We estimated the 17O influence from the δ18O value of N2O that we measured. We discussed the influence of Δ17O on δ15N values in a new section, “Quantification of the influence of Δ17O on δ15N values”. “The δ15N value of N2O measured by IRMS is calculated assuming Δ17O (=δ17O – 0.52 δ18O) is 0‰. However, since the Δ17O value of N2O may apparently increase the δ15N value, a correction was performed taking the Δ17O value into account. In this study, the Δ17O value of NO3− was not measured. On the other hand, the maximum δ18O value of N2O converted from NOx was 40‰ (containing laboratory and field experiments). Note that the δ18O value of N2O does not directly reflect the δ18O value of NOx because NOx obtains oxygen derived from water or H2O2 during oxidation in the recovery solution. The Δ17O values are generated only by mass transfer of O atoms from ozone to products during oxidation reactions [30]. In Eqs. 1-6, the Δ17O of O supplied during the process of NO and NO2 oxidizing NO3 is 0‰. Considering that NO and NO2 each receive two oxygen atoms, the Δ17O values become 2/3 or less (Eqs. 1–6). Further, it is assumed that the relationship between the Δ17O and δ18O values of NO2 is 0.36:1 (estimated by a straight line passing through the origin based on the values reported by Albertin et al. [12]). At this time, the maximum Δ17O value of N2O was 9.5‰ when the δ18O value was 40‰. The effect of this Δ17O value on the δ15N value was estimated using USGS34 and USGS35 with known Δ17O values. Based on the result of USGS35 measurements, when the Δ17O value is 21.56‰ [28], the apparent δ15N value would increase by 1.2‰. This result showed good agreement with the description by Yu and Elliott [16]. Assuming a linear relationship between Δ17O and δ15N values of USGS34 (Δ17O value: −0.3‰ [31]) and USGS35, the Δ17O value of 9.5‰ will increase the δ15N value by 0.5‰ at maximum. The overall measured 1σ uncertainty of δ15N(NOx) was ± 1.2‰ by combining the difference in the absorption efficiency of NO and NO2, the repeatability, and the consideration of Δ17O value.”

 We also added relevant references. Additionally, to explain the reaction, we added some reaction mechanisms in the Introduction part as follows: “In highly alkaline conditions, H2O2 produces various intermediate products that act as oxidants with H2O2 decomposition [20]. Free radicals generated due to H2O2 decomposition efficiently oxidize NO. It has been pointed out that particularly oxygen anions (O2−) produced at high pH may effectively oxidize NO [21]. On the other hand, NO and NO2 dissolve in NaOH solution, and the presence of H2O2 accelerates the oxidation of NO2 [18, 22]. The mechanism of the reaction of NOx with H2O2/NaOH is expressed as follows:”

H2O2 ⇆ OOH− + H+ (1)

H2O2 + OH−　⇆ HOO− + H2O (2)

H2O2 + HOO− → OH• + O2−• ＋ H2O (3)

O2−• + NO → ONOO− (4)

2NO2 + 2OH− → NO2− + NO3− + H2O (5)

NO2− + H2O2 → NO3− + H2O (6)

Since precision was changed from 1.0 to 1.2‰, we changed the error bars in Fig. 2.

Fig. Relationship between �17O and �18O values of NO2. Modified from Albertin et al. (2021).

References

Ho MC, Ong VZ, Wu TY, Potential use of alkaline hydrogen peroxide in lignocellulosic biomass pretreatment and valorization – A review. Renew. Sust. Energ. Rev. 2019; 112: 75–86, https://doi.org/10.1016/j.rser.2019.04.082.

Sun S, Zhang J, Sheng C, Zhong H, The removal of NO from flue gas by NaOH-catalyzed H2O2 system: Mechanism exploration and primary experiment, J. Hazard. Mater. 2022; 440: 129788, https://doi.org/10.1016/j.jhazmat.2022.129788.

Suzuki K, Niimi T, Yamamoto N, Shibata M, Saeki M, Ono A, et al. Rapid photometric method for the determination of the mass concentration of nitrogen monoxide and nitrogen dioxide. Anal. Chim. Acta 1994; 295: 135–141. https://doi.org/10.1016/0003-2670(94)80343-9. 

Michalski G, Savarino J, Böhlke JK, Thiemens M. Determination of the total oxygen isotopic composition of nitrate and the calibration of a Δ17Ο nitrate reference material. Anal. Chem. 2002; 74(19): 4989–4993. https://pubs.acs.org/doi/10.1021/ac0256282.n

Böhlke JK, Mroczkowski S J, Coplen T B, Oxygen isotopes in nitrate: New reference materials for 18O:17O:16O measurements and observations on nitrate-water equilibration. Rapid Commun. Mass Spectrom. 2003; 17: 1835−1846. https://doi.org/10.1002/rcm.1123.

Michalski, G, Scott Z, Kabiling M, Thiemens MH. First measurements and modeling of Δ17O in atmospheric nitrate, Geophys. Res. Lett. 2003; 30: 1870, doi:10.1029/2003GL017015, 16.

Line 170: Move the table footnotes describing superscripts "a" and "b" to immediately follow the table for clarity.

Reply: Corrected accordingly.

Line 185: If KMnO4 is a stronger oxidant than H2O2, speculate on why the solution with H2O2 exhibits higher collection efficiency.

Reply: Thank you for your question. In terms of redox power alone, KMnO4 is stronger than H2O2, but H2O2 produces OOH-, OH-, O2-, etc., at high pH. This information was added to the Introduction section. Therefore, the reason why the solution with H2O2 exhibited higher collection efficiency cannot be argued based solely on simple redox forces. Additionally, we could not find any evidence regarding the reaction of NH3 with KMnO4 or H2O2. Thus, we changed our conclusion to show that neither KMnO4 nor H2O2 showed a clear reaction with NH3. The new sentence was added on line 244 as follows: “However, since neither of the recovery solutions reacted by even 1% of the amount of ammonia added, it is hypothesized that a negligible reaction occurred with ammonia.”

---

## [Editor Report · Decision Letter 1]

26 Jan 2024

Low blank sampling method for measurement of the nitrogen isotopic composition of atmospheric NOx

PONE-D-23-29515R1

Dear Dr. Kamezaki,

We’re pleased to inform you that your manuscript has been judged scientifically suitable for publication and will be formally accepted for publication once it meets all outstanding technical requirements.

Kind regards,

Eva Stüeken, Ph.D.

Academic Editor

PLOS ONE

Additional Editor Comments (optional):

Dear Authors,

Thank you for carefully addressing all the comments. The manuscript is now ready for publication.

Best wishes,

Eva Stüeken
---

## [Editor Report · Acceptance letter]

21 Feb 2024

PONE-D-23-29515R1 

PLOS ONE

Dear Dr. Kamezaki, 

I'm pleased to inform you that your manuscript has been deemed suitable for publication in PLOS ONE. Congratulations! Your manuscript is now being handed over to our production team.

Kind regards, 

on behalf of

Dr. Eva Elisabeth Stüeken 

Academic Editor

PLOS ONE